# The Interleukin 6 Protein Level as well as a Genetic Variants, (rs1800795, rs1800797) Are Associated with Adverse Cardiovascular Outcomes within 10-Years Follow-Up

**DOI:** 10.3390/cells12232722

**Published:** 2023-11-28

**Authors:** Susanne Schulz, Selina Rehm, Axel Schlitt, Kerstin Bitter, Stefan Reichert

**Affiliations:** 1Department of Operative Dentistry and Periodontology, Martin-Luther-University Halle-Wittenberg, 06112 Halle (Saale), Germany; selinarehm@aol.com (S.R.); kerstin.bitter@uk-halle.de (K.B.); stefan.reichert@uk-halle.de (S.R.); 2Department of Cardiology, Paracelsus-Harz-Clinic Bad Suderode, 06485 Bad Sundered, Germany; axel.schlitt@pkd.de; 3Department of Medicine III, Medical Faculty, Martin-Luther-University Halle-Wittenberg, 06120 Halle (Saale), Germany

**Keywords:** interleukin 6 protein level, genetic variants, cardiovascular disease, prognostic factor

## Abstract

Background: Worldwide, cardiovascular disease (CVD) is the leading cause of premature death. The proinflammatory cytokine interleukin 6 (IL-6) is a essential marker of innate immunity that is considered to play an important proatherogenic role for cardiovascular disease. The aim of this study (substudy of ClinTrials.gov identifier: NCT01045070) was to evaluate IL-6 protein level and genetic variants (rs1800795, rs1800797) with respect to CV outcome (combined endpoint: myocardial infarction, stroke/transient ischemic attack, cardiac death, death according to stroke) among patients CVD within 10-years follow-up. Material and methods: Overall 1002 in-patients with CVD were included. IL-6 protein level was determined by electrochemiluminescence immunoassay (fasting, between 7 and 8 a.m.). Genetic analyses were carried out by single specific primer-polymerase chain reaction. Results: In survival analyses, IL-6 protein levels of ≥6.4 pg/mL (log-rank test: *p* = 0.034; cox regression: *p* = 0.032, hazard ratio = 1.29) and CC genotype of rs1800795 (log-rank test: *p* < 0.001, cox regression: *p* < 0.001, hazard ratio = 1.72) and AA genotype of rs180797 (log-rank test: *p* = 0.002, cox regression: *p* < 0.001, hazard ratio = 1.62) were associated with a poorer CV prognosis considering combined CV endpoint. Conclusion: This study was the first to investigate both elevated IL-6 levels and genetic variants for their prognostic value for adverse CV outcomes in CVD patients within the 10-year follow-up period.

## 1. Introduction

Cardiovascular disease (CVD) still represents the major contributor to overall global mortality [1]. Cardiovascular risk stratification is the benchmark for the use of effective preventive approaches that reflect optimal control of traditional and emerging risk factors. Experimental and clinical data indicates that inflammation participates fundamentally in atherogenesis [2]. Therefore, the assessment of inflammatory status has also been favored to improve cardiovascular risk stratification [3]. Recent research shows that inflammasomes can be considered as cellular signaling junctions of innate immunity, thereby controlling inflammatory pathways involved in atherosclerotic processes [4]. The importance of the inflammasome in atherosclerosis development, in addition to its effect on interleukin 1 family, is the downstream activation of the central signaling cytokine interleukin 6 (IL-6), which is associated with the enhanced release of C-reactive protein [5]. It is well established that IL-6 levels predict vascular events in primary and secondary prevention [3]. In clinical trials, high IL-6 levels have been shown to predict a range of cardiovascular (CV) outcomes [6,7]. Considering these results, the potential for IL-6 inhibition as a novel target for vascular protection could improve outcome for CVD patients [8]. A large-scale study (approx. 6.000 CVD patients), addressing this issue, is being conducted (ZEUS; IL-6 inhibitor: Ziltivekimab) [9].

There is existing evidence of a pro-inflammatory genetic pattern (e.g., IL-6 polymorphisms) which is recommended to be included in the risk profile of various inflammatory diseases [10]. Indeed, clinical studies have addressed the pathophysiological association of the IL-6 gene polymorphisms to CVD [11]. The focus of these studies was on the functional single nucleotide polymorphism (SNP) rs1800795, which is located in the promoter of the IL-6 gene [11,12]. In a meta-analysis, comprising of 74 studies including 86,229 subjects, the CC genotype of rs1800795 was associated with increased CV risk [11]. The identification of stable genetic markers (e.g., in IL-6 gene), which could be referred to CV outcome, would be clinically of substantial interest, too. However, data regarding the prognostic value of SNP rs1800795 regarding CVD are controversial. Therefore, it is supposed that affecting factors, e.g., patients characteristics, influence the study results [13,14]. 

The aim of the present study was to investigate the influence of SNPs rs1800795 and rs1800797 in IL-6 gene and IL-6 serum level on CV prognosis. Here for, a cohort of CVD patients was studied during a follow-up period of 10 years with regard to a recurrence of CV events (CV death, death from stroke, MI, and stroke/transient ischemic attack (TIA)). This study approach was designed to substantiate and extend the 3-year follow-up results, which emphasized the CC genotype (rs1800795) as a prognostic marker for CV outcome [15]. 

## 2. Materials and Methods

### 2.1. Study Design

The study presented here is a substudy of “Periodontitis and Its Microbiological Agents as Prognostic Factors in Patients with Coronary Heart Disease” (ClinicalTrials.gov identifier: NCT01045070, 21 October 2008). The ethics committee of the Martin Luther University Halle-Wittenberg approved the investigation on the 18 July 2007. Written informed consent was given by all patients. The ethical guidelines of according to “Declaration of Helsinki” including its amendment in “Tokyo and Venice” were applied. The longitudinal cohort study was performed with a 10-year follow-up period.

### 2.2. CV Patients

Patient details and their clinical assessments have been comprehensively described in Reichert et al., 2016. [16]. Briefly illustrated: From October 2009 to February 2011, 1002 cardiovascular patients admitted to the Department of Cardiac and Thoracic Surgery and the Department of Internal Medicine III, Martin Luther University Halle-Wittenberg were prospectively included. All patients underwent coronary angiography. 

Inclusion criteria: All patients were over the age of 18. They had to have at least a 60% stenosis of a main coronary artery. Stenosis was proven by current or previous coronary artery bypass surgery (CABG), by coronary angiography and current or previous percutaneous coronary intervention (PCI). 

Exclusion criteria: periodontological treatment (scaling and root planing) within the last 6 month, antibiotic treatment (last 3 months), pregnancy, inability to give consent to the study, current abuse of alcohol or drugs. 

After confirmation of inclusion and exclusion criteria anamnestic date were collected on the day after inpatient admission. Between 7–8 a.m. fasting blood withdrawal was performed. For assessment of biochemical factors standard methods of the Central Laboratory of the Medical Faculty of the Martin Luther University Halle Wittenberg were applied. Additionally, baseline variables (gender, age, smoking, body mass index) patients’ disease history (e.g., diabetes mellitus, myocardial infarction, stroke, transient ischemic attack (TIA), hypertension, peripheral arterial disease, dyslipoproteinemia) were obtained. 

60 patients were not included in the final IL-6 baseline study population due to missing blood sampling (baseline: *n* = 942).

742 CV patients were assessed (November 2020–January 2023) in the 10-year follow-up. The predefined combined CV endpoint consisted of MI, CV death, stroke/TIA, death from stroke. For acquiring follow-up data a standardized questionnaire was sent by post. if the questionnaire was not returned, a telephone survey was conducted (patient, patient’s relatives). In the event of patient’s death the attending physician was contacted. If required, civil registration offices were ask about address or date of death.

### 2.3. Interleukin 6 Analysis

The genetic analyses (rs1800795, rs1800797) were carried out with PCR-SSP (sequence specific oligonucleotides) using the CYTOKINE Genotyping array CTS-PCR-SSP kit (Collaborative Transplant Study, Department of Transplantation Immunology of the University Clinic of Heidelberg, Heidelberg, Germany). Genotypes, as well as alleles and haplotypes were assessed. 

Interleukin 6 serum levels were assessed in the Central Laboratory of the Medical Faculty of the Martin Luther University Halle Wittenberg applying electrochemiluminescence immunoassay (ECLIA) using a Cobas c 701 module (Roche Diagnostics International AG, Rotkreuz, Switzerland; reference range: ≤6.4 pg/mL) 

### 2.4. Statistical Evaluation

Statistical analyses were carried out using SPSS 25.0 (SPSS Inc., Chicago, IL, USA) and *p* values ≤ 0.05 were considered to be significant. Kolmogorov–Smirnov test and Shapiro–Wilk test were applied for testing for normal distribution (continuous data: metric demographic, clinical, and serological data). Non-normally distributed data were reported as median, 25th/75th interquartiles and analysed using Mann–Whitney U test. Categorical variables were documented as percentage, applying the chi-squared test (including Fisher’s exact test: expected values < 5) for statistical analyses. Mann-Whitney U tests and chi-squared tests were used for assess the association of demographic, clinical, serological and genetic data with elevated IL-6 level (Table 1). Furthermore, Mann-Whitney U test and chi-squared test were performed to evaluate the impact of IL-6 level and rs1800795 and rs1800797 on combined endpoint (Table 2 and Table 3). 

In survival analyses, carried out by Kaplan–Meier plots, Log-Rank test (univariate analyses), and Cox regression (adjusted Hazard ratios; HR), the impact of IL-6 level and genetic variants rs1800795 and rs1800797on CV outcome were evaluated. In order to generate Kaplan–Meier plots, originally metric IL-6 values were dichotomized regarding clinical aspects (≤6.4 pg/mL vs. >6.4 pg/mL). Adjusted HRs were generated for IL-6 level and genetic variants (rs1800795, rs1800797) with Cox regression with respect to known risk factors for CVD. 

## 3. Results

### 3.1. Clinical Characterisation of CV Patients

A total of 1002 patients with cardiovascular disease were recruited at baseline. 942 patients were included in the final IL-6 baseline study population. Of these patients, 742 completed the 10-year follow-up. The combined endpoint (CV deaths, deaths from stroke, MIs, and strokes/TIAs) was met by 315 (42.5%) of the patients (flowchart of study design: Figure 1). 

The following baseline calculations included 942 patients from whom IL-6 serum levels were available. Baseline characteristics (demographical and anamnestic parameters, genotype distribution, medical history) according to standard (≤6.4 pg/mL) and pathological (>6.4 pg/mL) IL-6 levels are displayed in Table 1. 

### 3.2. IL-6 Serum Level as Prognostic Markers for Adverse CV Outcome

In statistical analyses not considering time, elevated IL-6 levels were shown to be associated with the occurrence of a combined endpoint (*p* = 0.017) (Table 2). 

To reflect survival time, IL-6 levels were dichotomized. The internal laboratory reference value (Central Laboratory, Medical Faculty, Martin-Luther-University Halle-Wittenberg) was considered as parameter for dichotomization (≤6.4 pg/mL vs. >6.4 pg/mL). In Kaplan-Meier survival curve including log-rank test, an elevated IL-6 level is associated with an adverse CV outcome within 10-year follow-up (*p* _log rank_ = 0.034, HR: 1.28) (Figure 2). 

### 3.3. SNPs rs1800795, rs1800797 in IL-6 Gene as Prognostic Markers for Adverse CV Outcome

The two SNPs studied in this work (rs1880795, rs1800797) were examined for the occurrence of linkage disequilibrium. The haplotype analysis identified strong linkage disequilibrium (logarithm of odds (LOD) > 10) within the study population (Figure 3). 

In addition, it was evaluated whether the two SNPs rs1800795 and rs1800797 were associated with the level of Il-6. For both SNPs no genetic dependency with interleukin 6 level was assessed (Figure 4). 

The SNPs rs1800795 and rs1800797 in the IL-6 gene were analyzed for its prognostic value according to CV outcomes. In contingency analysis, a worse CV outcome was associated with CC genotype of rs1800795 (*p* = 0.014) as well as AA genotype of rs1800797 (Table 3).

Survival time was included in subsequent evaluations of the prognostic potential of both SNPs in the IL-6 gene on CV outcome. In the Kaplan-Meier survival curve and log-rank test, CC genotype of SNP rs180795 was associated with worse CV prognosis (*p* _log rank_ < 0.001; Figure 5). 

A similar result was obtained analyzing SNP rs1800797 regarding its impact on CV survival. Patients carrying AA genotype had a poorer CV outcome in 10-year follow-up (*p* _log rank_ = 0.002; Figure 6). 

### 3.4. Adjustment of Cofactors in Multivariate Analysis 

In Cox regression analysis, adjusted Hazard ratios were assessed. Established CV risk markers e.g., male gender, age, BMI, smoking, diabetes mellitus, hypertension, and dyslipoproteinemia were considered in the calculation. An elevated IL-6 level (HR: 1.29, 95% CI: 1.02–1.62), CC genotype of rs1800795 (HR: 1.72, 95% CI: 1.33–2.23) and AA genotype of rs1800797 (HR: 1.62, 95% CI: 1.25–2.11) could be definite as prognostic factors for adverse CV outcome in the complex risk model. Furthermore, factors associated with an poorer CV prognosis were older age, male gender, and the presence of diabetes mellitus (Table 4).

## 4. Discussion

In this longitudinal cohort study, the impact of interleukin 6 regarding genetic variants (rs1800795, rs1800797) and circulating serum level on future CV events was investigated. For this objective, a cohort of patients with angiographically proven CVD was enrolled and followed over a 10-year period.

### 4.1. Elevated IL-6 Serum Level as Risk Marker for Adverse CV Prognosis

There is increasing evidence that the cardiovascular development and prognosis in CV patients is related to persistent vascular inflammation, linked, among others, to interleukin 6 [17,18]. As shown in previous studies, the present investigation also confirmed that elevated IL-6 serum levels are associated with an adverse CV prognosis even after 10 years [3]. In the meta-analysis by Antonopoulos et al., the added prognostic value (Δ(c-index%)) for major adverse clinical outcome events was shown to be 1.7 for increased Il-6 serum level (*n* = 21,512). In line with this meta-analysis, our results indicate that measurement of circulating interleukin-6 levels provides further prognostic information in addition to traditional clinical risk models. In the present study, the impact of elevated IL-6 level on CV outcome was also significant even when classical CV risk factors (age, male gender, body mass index, smoking, history of diabetes mellitus, hypertension, hyperlipoproteineamia) were taken into account (hazard ratio: 1.29, *p* = 0.032). Therefore, the implementation of biomarkers of vascular inflammation, e.g., Il-6, is a rational strategy for CV risk stratification in future. 

Furthermore, inflammatory markers, such as IL-6, are recommended as potential targets for effective CV treatment. For example, the CANTOS trial provided indication that IL-6 may have a pivotal impact in CV prognosis. In this study, IL-1β antibody treatment reduced the risk of major adverse cardiovascular event rates accompanied by a lowered Il-6 level [19]. Correspondingly, IL-6 signaling pathway is receiving increased attention as a potential target for CVD intervention [20]. A pilot smallscale study was conducted to assess the effect of an IL-6 antagonist (Ziltivekimab) on biomarkers of inflammation relevant to CVD (RESCUE: double-blind, randomised, placebo-controlled, phase 2 trial, *n* = 264) [8]. Based on this, the promising protective effect of Ziltivekimab will be investigated in a large-scale study in which the patients are also stratified regarding left ventricular ejection fraction as a marker of heart failure (ZEUS; approx. 6000 patients) [9]. 

### 4.2. SNPs in IL-6 Gene as Prognostic Markers for Adverse CV Outcome

Stable markers, such as genetic characteristics, are favorable as risk as well as prognostic markers for multifactorial diseases, like CVD. Studies on the association of SNPs with CVD have also been conducted with regard to genetic variants in the IL-6 gene [11]. 

In the meta-analysis conducted by González-Castro et al. the importance of this SNP was investigated based on 74 studies including 86,229 CVD patients [11]. In this study, the CC genotype of rs1800795 was associated with CVD in the entire study group (OR = 1.11, CI 95% = 1.03–1.19, *p* value = 0.002). Furthermore, it was shown that the contribution of SNP rs1800795 to CVD is subject to ethnic differences and CV diagnosis (coronary artery disease, ischemic stroke, myocardial infarction, and peripheral arterial occlusive disease). The strongest association was observed for CC genotype and coronary artery disease (OR = 1.50, CI 95% = 1.28–1.76, *p* value < 0.0001). 

In the present study, however, the focus was not on the significance of genetic variants of IL-6 for the incidence but on the prognosis after CV event. We could associate the CC genotype of rs1800795 and AA genotype of rs1800797 with adverse CV events in 10-year follow-up (rs1800795: log-rank test, *p* < 0.001, rs1800797: log-rank test, *p* = 0.002). As there is no equivalent data for the SNP rs1800797, the results cannot be compared with other studies. Previous data on the dependence of the SNP rs1800975 on the prediction of CV outcome are controversial. Results of Máchal et al. were comparable to the data of the present study [14]. They demonstrated higher mortality of CC genotype carriers (HR = 3.79; 95% CI = 1.78–8.10; *p* < 0.001). Although they included patients with at least 50% stenosis of one of the main coronary arteries (similar to the present study), the defined endpoint was very different (overall mortality vs. CV endpoint: CV death, death from stroke, MI, and stroke/attack (TIA)). In another study by Sukhinina et al., no association of this SNP to cardiac death, recurrent myocardial infarction, recurrent hospitalization with unstable angina or stroke, coronary bypass surgery, percutaneous transluminal coronary angioplasty was assessed in patients suffering from myocardial infarction in two-year follow-up [13]. In summary, because of differences in inclusion criteria and the defined endpoint, no consistent conclusions can be drawn regarding prognostic value of rs1800795. 

### 4.3. Interaction of Genetic Variants and IL-6 Serum Level 

It was demonstrated that the level of IL-6 expression is influenced by genetic predisposition, among other factors [21]. In vitro studies have focused on SNPs located in the promoter of the IL-6 gene, especially rs1800795 [21,22]. However, these studies are controversial. In reporter assays a decrease in IL-6 protein expression was associated with C allele of rs1800795 [21,23]. In contrast, in a further in vitro study, a high IL-6 expression was assessed for C allele in human fibroblasts [24]. In addition, evidence was found that the genetic influence on IL-6 expression is dependent on the cell type [24]. 

In clinical trials an increase in IL-6 expression was observed in CVD patients carrying CC genotype or C allele of rs1800795 [11,25]. Increased IL-6 expression associated with the CC genotype/C allele would be consistent with the adverse CV outcome for IL-6 level and rs1800795 described in the present study (IL-6 level: Figure 2; CC genotype: Figure 5). However, in the present study no significant correlation between IL-6 SNP rs1800795 [26] or rs1800797 and IL-6 serum level was observed. In the overall group, CC genotype carriers of rs1800795 as well as AA carriers of rs1800797 were shown to have slightly lower IL-6 levels than CG + GG or AG + GG genotype carriers, respectively (Figure 4). This discrepancy between the different studies regarding could be due to the fact that patients of varying ethnic origins were evaluated. Furthermore, it is possible that a direct association between genetic predisposition and phenotypic expression is covered by other pathways in the complex human metabolic processes. Therefore, in vitro cell culture experiments should be focussed on when investigating the genetic influence on expression.

### 4.4. Limitation of the Study

The study design of the present investigation has been a longitudinal cohort study. The study was performed to assess assumptions about the potential prognostic value of genetic variations in IL-6 and IL-6 serum levels in terms of adverse CV outcomes during a 10-year follow-up period. Since direct association between SNP rs1800795 and IL-6 level could not be demonstrated in the present study, causality based on the results cannot be assured. Due to the lost to follow-up rate in the present study, the statistics should only be evaluated with appropriate consideration. The results of this study are valid for patients with angiographically proven CVD of Central Germany (Caucasian origin). The results cannot be generalized to the overall population or other patient cohorts.

## 5. Conclusions

The present study emphasis the importance of an elevated IL-6 serum level (≥6.4 pg/mL) and the CC genotype of rs1800975 for CV outcome in the present cohort of CV predisposed patients. These assumptions were confirmed to be independent of the classical cardiovascular risk factors. However, an inclusion of IL-6 serum level and genetic variant rs1800975 in the CV risk profile is currently premature and requires further clinical investigations. Nevertheless, CV patients with elevated IL-6 serum levels and the presence of CC genotype are at particular risk for adverse CV events and should be monitored at frequent recall intervals. 

## Figures and Tables

**Figure 1 cells-12-02722-f001:**
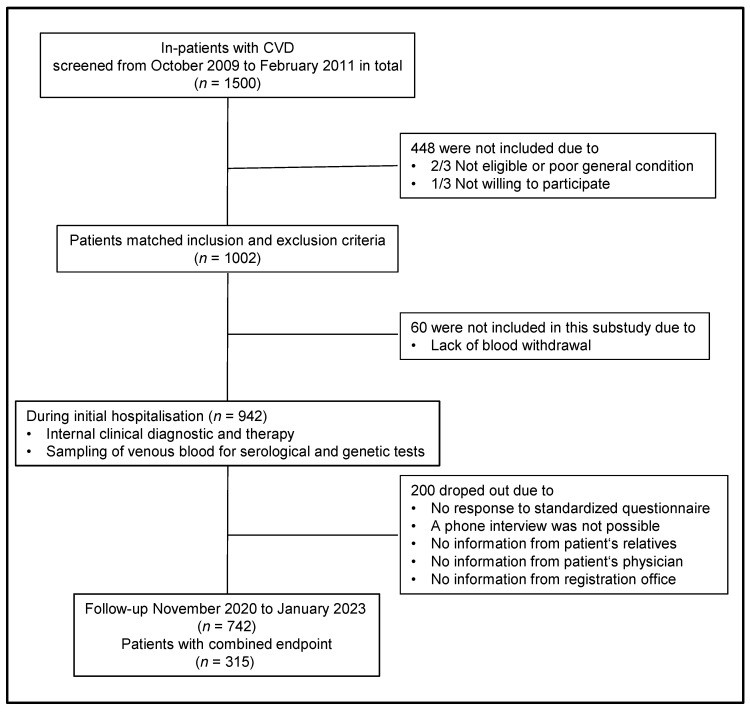
Study design and flow.

**Figure 2 cells-12-02722-f002:**
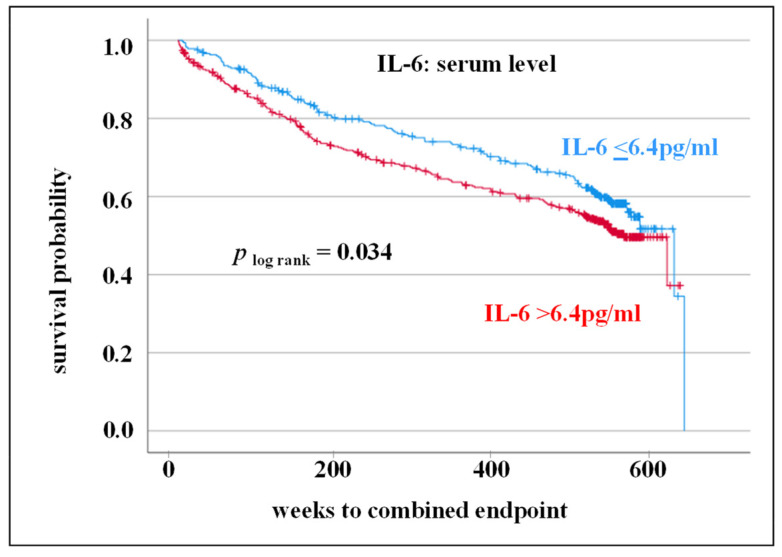
Kaplan-Meier survival curve (including log-rank test) for circulating interleukin 6 (IL-6) levels (≤6.4 pg/mL vs. >6.4 pg/mL) with an impact on combined endpoint (myocardial infarction, cardiovascular death, stroke/transient ischemic attack, death from stroke).

**Figure 3 cells-12-02722-f003:**
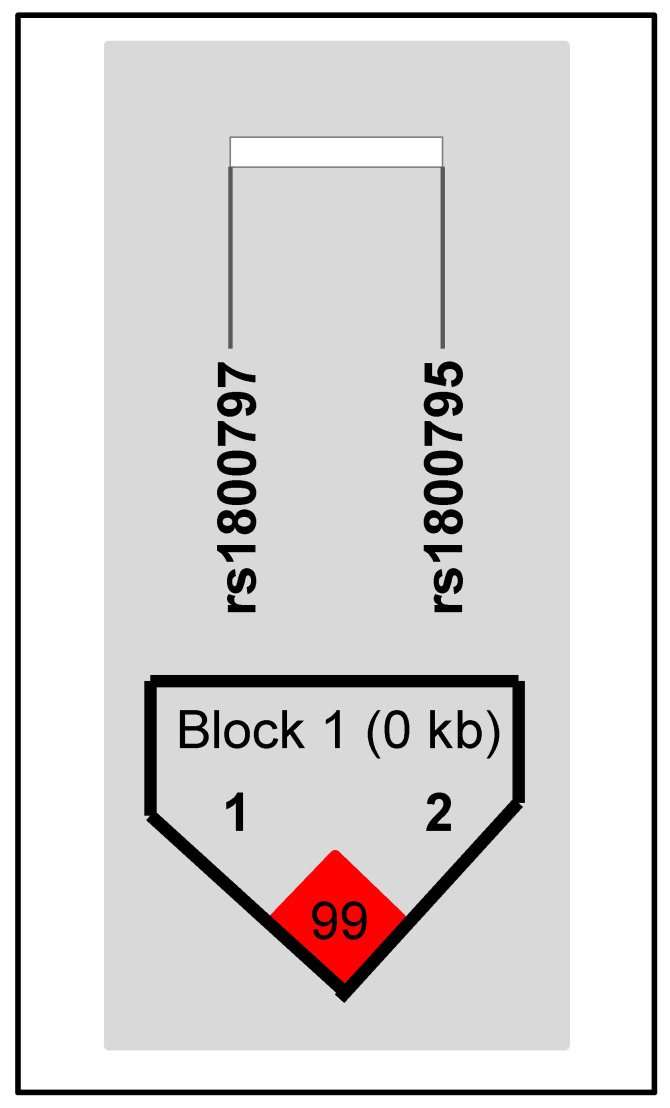
Haplotype block structure determined using Haploview 4.2. Square: linkage disequilibrium and LOD score between interleukin 6 (IL-6) single nucleotide polymorphisms rs1800795 and rs1800797; Linkage disequilibrium is displayed as pairwise D’ values in the red square.

**Figure 4 cells-12-02722-f004:**
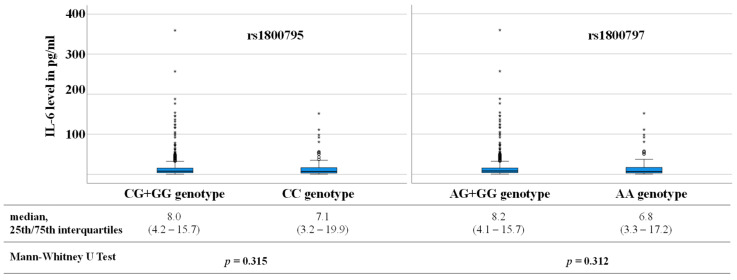
Genetic influence (rs1800795, rs1800797) on interleukin 6 (Il-6) level. Outliers are marked with an asterisk.

**Figure 5 cells-12-02722-f005:**
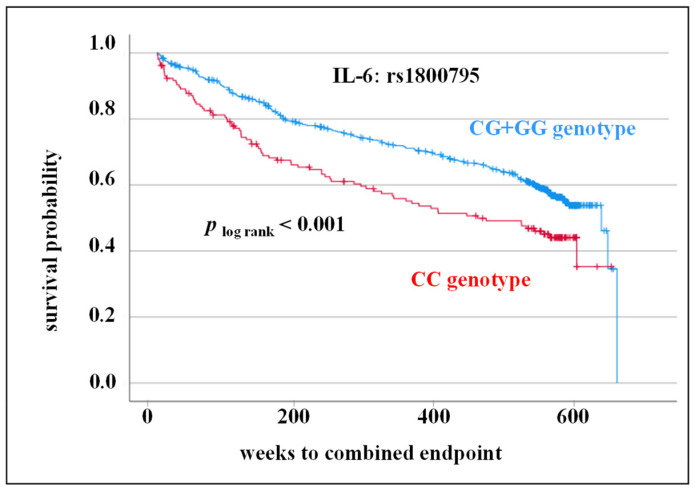
Kaplan-Meier survival curve including log-rank test for the composite endpoint (myocardial infarction, cardiovascular death, stroke/transient ischemic attack, death from stroke) regarding genotype distribution of rs1800795 in interleukin 6 (IL-6) gene.

**Figure 6 cells-12-02722-f006:**
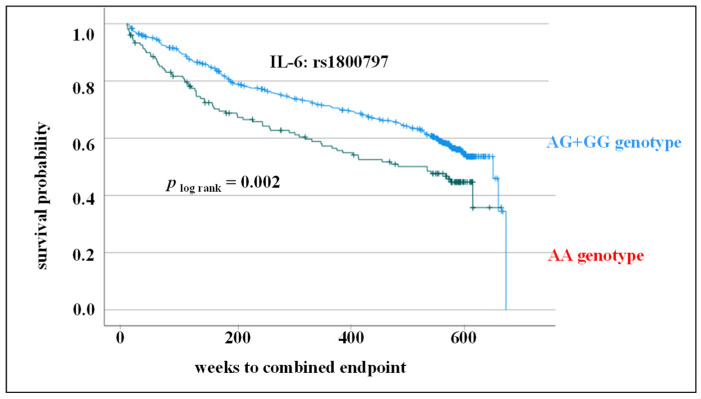
Impact of rs18007979 in interleukin 6 gene on survival in 10-year follow-up: Kaplan-Meier survival curve including log-rank test for the composite cardiovascular endpoint (myocardial infarction, cardiovascular death, stroke/transient ischemic attack, death from stroke).

**Table 1 cells-12-02722-t001:** Baseline characteristics of cardiovascular patients with respect to interleukin 6 (IL-6) serum level (within the reference range: ≤6.4 pg/mL vs. above medical reference range: >6.4 pg/mL). Non-parameterised variables were assessed by Mann-Whitney U test and plotted as median (IQR: 25th/75th-interquartiles). Categorical variables are displayed as a percentage and evaluated using chi-squared test. (IQR: interquartile range; MI: myocardial infarction; TIA: transient ischemic attack).

Characteristics	All Patients (*n* = 942)	IL-6 ≤ 6.4 pg/mL (*n* = 422)	IL-6 > 6.4 pg/mL (*n* = 520)	*p*-Value
Demographical and anamnestic parameters				
Age, years (median; 25/75 IQR)	69.1 (60.4/74.7)	68.9 (60.0/74.2)	69.3 (60.6/75.0)	0.331
Female gender (%)	26.7	30.3	23.9	0.058 ^1^
Current smoking (%)	10.6	11.1	10.3	0.790 ^1^
Body mass index, kg/m^2^ (median; 25/75 IQR)	28.1 (25.3/30.7)	28.1 (25.2/30.5)	28.1 (25.4/30.7)	0.560
History of				
Diabetes mellitus (%)	34.5	33.7	35.1	0.763 ^1^
Hypertension (%)	87.6	88.5	86.9	0.567 ^1^
MI (%)	38.9	39.3	38.7	0.916 ^1^
Stroke/TIA (%)	12.5	13.3	11.9	0.652 ^1^
Peripheral artery disease (%)	9.7	9.0	10.3	0.645 ^1^
Hyperlipoproteinemia (%)	59.2	61.0	57.8	0.416 ^1^
Polymorphisms in IL-6 gene				
rs1800795				
CC genotype (%)	21.3	23.0	20.0	
CG + GG genotype (%)	78.7	77.0	80.0	0.365 ^1^
rs1800797				
AA genotype (%)	20.3	22.4	18.8	
AG + GG genotype (%)	79.7	77.6	81.3	0264 ^1^

^1^ Yates correction.

**Table 2 cells-12-02722-t002:** Interleukin 6 (IL-6) level of cardiovascular patients regarding combined endpoint (myocardial infarction, cardiovascular death, stroke/transient ischemic attack, death from stroke) in 10-year follow-up). (IQR: 25th/75th-interquartiles).

	All Patients (*n* = 742)	Without Combined Endpoint (*n* = 427)	With Combined Endpoint (*n* = 315)	*p*-Value
IL-6 (pg/mL), (median; 25/75 IQR)	7.8 (3.8/15.9)	7.2 (3.4/13.9)	8.7 (4.3/18.5)	0.017 ^1^

^1^ Mann-Whitney U test.

**Table 3 cells-12-02722-t003:** Association of single nucleotide polymorphisms (SNPs), rs1800795 and rs1800797 in the interleukin 6 gene and combined endpoint (myocardial infarction, cardiovascular death, stroke/transient ischemic attack, death from stroke, in 10-year follow-up).

	All Patients (*n* = 742)	Without Combined Endpoint (*n* = 427)	With Combined Endpoint (*n* = 315)	*p*-Value
rs1800795				
CC genotype (%)	21.2	17.9	25.6	
CG + GG genotype (%)	78.8	82.1	74.4	0.014 ^1^
rs1800797				
AA genotype (%)	20.2	17.4	24.1	
AG + GG genotype (%)	79.8	82.6	75.9	0.033 ^1^

^1^ Yates correction.

**Table 4 cells-12-02722-t004:** Cox regression for evaluation of the prognostic impact of an increased interleukin 6 (IL-6) level and CC genotype of IL-6 SNP rs1800795 (a) and AA genotype of rs1800979 (b) on the incidence of combined endpoint (myocardial infarction, cardiovascular death, stroke/transient ischemic attack death from stroke) taken classical cardiovascular risk factors as covariates into consideration. (CI: confidence interval, BMI: body mass index).

	Regression Coefficient	Standard Error	*p*-Value	Hazard Ratio	95% CI	
					Upper	Lower
(**a**)						
**CC genotype (rs1800795)**	0.545	0.131	**<0.001**	1.72	1.33	2.23
**IL-6 level > 6.4 pg/mL**	0.251	0.117	**0.032**	1.29	1.02	1.62
Age	0.042	0.007	<0.001	1.04	1.03	1.06
Male gender	0.329	0.135	0.015	1.39	1.07	1.81
BMI	−0.014	0.014	0.346	0.99	0.96	1.02
smoking	0.308	0.218	0.156	1.36	0.89	2.08
Diabetes mellitus	0.552	0.122	<0.001	1.74	1.37	2.21
Hypertension	0.143	0.192	0.454	1.15	0.79	1.68
Hyperlipoproteinemia	−0.119	0.119	0.315	0.89	0.70	1.12
(**b**)						
**AA genotype (rs1800797)**	0.484	0.134	**<0.001**	1.62	1.25	2.11
**IL-6 level > 6.4 pg/mL**	0.248	0.117	**0.033**	1.29	1.02	1.62
Age	0.042	0.007	<0.001	1.04	1.03	1.06
Male gender	0.327	0.135	0.016	1.39	1.06	1.81
BMI	−0.013	0.014	0.366	0.99	0.96	1.02
smoking	0.299	0.218	0.170	1.35	0.88	2.07
Diabetes mellitus	0.538	0.122	<0.001	1.71	1.35	2.18
Hypertension	0.137	0.192	0.474	1.15	0.79	1.67
Hyperlipoproteinemia	−0.110	0.119	0.355	0.90	0.71	1.13

## Data Availability

The authors are available to provide all data on request.

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
