# Peer review of "The Interleukin 6 Protein Level as well as a Genetic Variants, (rs1800795, rs1800797) Are Associated with Adverse Cardiovascular Outcomes within 10-Years Follow-Up"

_cells, 2023, doi:10.3390/cells12232722_

Round 1
Reviewer 1 Report
Comments and Suggestions for Authors
Comments:
Title: The Interleukin 6 protein level as well as genetic variants (rs1800795, rs1800797) are predictors for adverse cardiovascular outcomes within 10-years follow-up.
Authors: Susanne Schulz, Selina Rehm, Axel Schlitt, Kerstin Bitter, and Stefan Reichert.
Summary: no comments.
Introduction: The purpose of the study was to investigate the association between the rs 1800795/rs 1800797 polymorphisms of the IL6 gene and the plasma level of IL6 with the recurrence of MACE in individuals with coronary ASCVD.
Material and Methods: original population 1,002 individuals with angiographically proven ASCVD. Of this cohort, 792 were evaluated (21% not evaluated), and finally 742 individuals (26% not evaluated) were analyzed with genotype and IL6 level.
Statistics: no comments.
Results: with a cut-off point of 6.4pg/ml of IL6 in plasma, a significant correlation was reported between IL6 level and the incidence of MACE; likewise, a significant correlation was reported between the SNP rs1800795 and the incidence of MACE.
Suggestions:
a) correct lines 170 and 188 and in table 2, and line 174: rs1800795 (says 1880795) and rs 1800797 (says 180797).
b) specify that the relationship between the SNP rs 1800795 and the serum level of IL6 was not analyzed and the reason for that.
Discussion: the authors postulate that based on the results there is a link between the SNP-IL6 rs 1880795 and the IL6 value with the incidence of MACE, however, since the study does not present in the results section the association between SNP rs 1800795 and the value of IL6 in plasma and its linear link with the incidence of MACE, it is suggested:
a) propose association and specify that since linearity cannot be demonstrated between SNP rs 1800795, IL6 level and MACE incidence, causality based on the results cannot be assured.
b) clarify that the ZEUS study is different from the HERMES study.
c) discuss that genetic biomarkers and cytokines could be considered not only based on the association, among other characteristics they should demonstrate that they significantly improve the performance of traditional algorithms, and it is still necessary to report safety, improved outcomes and cost-efficacy before proposing them for clinical use.
d) recognize in the weaknesses section of the study that it did not include 26% of the original cohort, a sensitivity analysis would have been desirable.
Conclusions: I suggest modulating them according to my suggestions.
Bibliography: no observations, consistent with the text.
Reviewer 2 Report
Comments and Suggestions for Authors
The present study by Schulz et al. evaluated IL- 6 protein level and genetic variants (rs1800795, rs1800797) with respect to CV outcome among patients CVD within 10-years follow-up and came up with conclusions that CV patients with elevated IL-6 serum levels and the presence of CC genotype are at particular risk for adverse Cardiovascular events and should be monitored at frequent recall intervals. Although, the study seems to be interesting, however, I have some major comments as follows:
Major comments
1. The novelty of the study or what is something new that this study is offering is not clearly stated in abstract of the study which is indeed recommended.
2. Why patients with number of teeth > 4 were chosen for this study?
3. In line with point 2, can authors describe why did they chose specific inclusion and exclusion criteria for study samples as these criteria seem to be in context of their bigger study which is “Periodontitis and Its Microbiological Agents as Prognostic Factors in Patients with Coronary Heart Disease”, of which current study samples are derived from. It is hard to correlate how dental features can correlate with CV outcomes for present study related question.
4. What were the major reasons for 21% dropout rate in sample cohort?
5. How did authors choose the standard IL-6 levels as 6.4pg/ml. Authors should provide a reference study for the same.
6. It will be great to state the biological question followed by why a specific statistical test was applied to answer that question instead of merely mentioning the name of the statistical test. This will be helpful for readers to understand why a specific statistical test was performed.
7. Having a supplementary flowchart displaying sample numbers would be beneficial. This flowchart should include details such as the initial sample numbers, dropped-out samples, the count of individuals with demographic and anamnestic data, those who had IL-6 measurements, individuals genotyped for IL-6 polymorphism, and the number of cases with and without combined endpoints in each category. This will help maintain consistency in sample numbers across various comparisons presented in the tables.
8. Authors emphasis the inclusion of an elevated IL-6 serum level (>6.4pg/ml) and the CC genotype of rs1800975 in the cardiovascular prognosis profile for CV predisposed patients. These assumptions were confirmed to be independent of the classical cardiovascular risk factors. However, what about cardiovascular health protective factors such as exercise or healthy diet? Do authors consider including information regarding these protective factors of samples in the model to see do they have any impact on relating elevated IL-6 serum level (>6.4pg/ml) and the CC genotype of rs1800975 on CV outcomes?
Minor comments
1. In line number 119, there seems to be a typo error. The line says – “p values > 0.05 were considered to be significant”. However, if I correctly understand, authors want to say - p values < 0.05 were significant”. Please check.
2. Table 3 should be renamed as Table 2 and Table 2 should be renamed as Table 3 as per their order in manuscript.
3. Figure 2 has a very poor resolution. I encourage authors to add a better resolution figure 2.
Author Response
Dear Reviewer,
Thank you very much for your invaluable suggestions for revising our manuscript. We introduced your comments and critical remarks in the revised manuscript.
Here, we want to explain how the manuscript was revised. The changes are indicated using blue color.
Major comments
- The novelty of the study or what is something new that this study is offering is not clearly stated in the abstract of the study which is indeed recommended.
Thank you very much for this helpful advice. The abstract was revised and it was stated that this study was the first to investigate both elevated IL-6 levels and genetic variants for their prognostic value for adverse CV outcomes in CVD patients within the 10-year follow-up period.
- Why patients with number of teeth > 4 were chosen for this study?
The study presented here is a substudy of the study “Periodontitis and Coronary Heart Disease” (ClinTrials.gov identifier: NCT01045070). As the number of teeth is highly relevant for the main study, this factor was considered as an inclusion criterion in this study (For microbiological subgingival examinations, which were carried out in the main study, it was necessary to have 4 own teeth.). However, the reviewer is of course right that the inclusion criterion "number of teeth" is not relevant for the data analysed in this substudy. For this reason, it is not necessary to consider the number of teeth for the IL-6 analyses and could be omitted here.
- in line with point 2, can authors describe why did they chose specific inclusion and exclusion criteria for study samples as these criteria seem to be in context of their bigger study which is “Periodontitis and its microbiological agents as prognostic factors in patients with coronary heart disease”, of which current study samples are derived from. It is hard to correlate how dental features can correlate with CV outcomes for present study related questions.
The answer to point 2 also applies here. The inclusion of the "number of teeth" is not relevant for the study presented here.
- What were the major reasons for 21% dropout rate in sample cohort?
Within the present study, 1002 patients were enrolled in the investigation (ClinTrials.gov identifier: NCT01045070). 60 patients were not included in the final IL-6 baseline study population due to missing blood sampling. Of the 942 patients included in this IL-6 study at baseline, 941 patients completed the 1-year follow-up, 895 patients completed the 3-year follow-up and 742 completed the 10-year follow-up.
Follow-up: Firstly, all patients were contacted by post and asked to complete a questionnaire. If this procedure failed, the patients were interviewed by telephone. If follow-up information could not be obtained by questionnaire or telephone interview, we contacted first relatives or patient’s physician and second civil registration offices and requested information about current address or date of death. The missing patients in the follow-up droped out due to: a) no response to standardized questionnaire, b) a telephone interview was not possible, c) No information from patient’s relatives, d) No information from patient’s physician e) no information from registration office.
- How did authors choose the standard IL-6 levels as 6.4pg/ml. Authors should provide a reference study for the same.
Interleukin 6 serum levels were assessed in the Central Laboratory of the Medical Faculty of the Martin Luther University Halle Wittenberg applying electrochemiluminescence immunoassay (ECLIA) using a Cobas c 701 module (Roche Diagnostics International AG, Rotkreuz, Switzerland). The analysis at that time (October 2009 to February 2011) used the then valid reference values according to the Central Laboratory of the Medical Faculty of the Martin Luther University Halle Wittenberg (>6.4pg/ml). The reference values were included in the manuscript.
- It will be great to state the biological question followed by a specific statistical test was applied to answer that question instead of merely mentioning the name of the statistical test. This will be helpful for readers to understand why a specific statistical test was performed.
Thank you very much for your advice. The method section was revised in this regard:
Mann-Whitney U tests and chi-squared tests were used for assess the association of demographic, clinical, serological and genetic data with elevated IL-6 level (table 1). Furthermore, Mann-Whitney U test and chi-squared test were performed to evaluate the impact of IL-6 level and rs1800795 on combined endpoint (tables 2, 3).
In Survival analyses, carried out by Kaplan–Meier plots, Log-Rank test (univariate analyses), and Cox regression (adjusted Hazard ratios; HR), the impact of IL-6 level and genetic variants rs1800795 on CV outcome was evaluated. In order to generate Kaplan–Meier plots, originally metric IL-6 values were dichotomized regarding clinical aspects (<6.4pg/ml vs. >6.4pg/ml). Adjusted HRs were generated for IL-6 level and genetic variants (rs1800795) with Cox regression with respect to known risk factors for CVD.
- Having a supplementary flowchart displaying sample numbers would be beneficial.
For a better overview a flowchart displaying the detailed sample numbers is given in figure 1 of the revised manuscript.
- However, what about cardiovascular health protecting factors such as exercise or healthy diet? Do authors consider including information regarding these protective factors of samples in the model to see do they have any impact on relating elevated Il-6 serum level (>6.4pg/ml) and the CC genotype of rs1800795 on CV outcome?
Unfortunately, when the study was designed, not enough attention was paid to health-supporting factors such as a healthy diet and physical activity or other potentially influencing parameters, including socioeconomic factors, education level or stress management. If the authors were to plan the study again, greater consideration would be given to these particular factors.
Minor comments
- In line number 119, there seems to be a typo error. The line says -“p values >0.05 were considerd to be significant.
Thank you very much for this information. The sentence has been corrected.
- Tabel 3 should be renamed as Table 2 and Table 2 should be renamed as Table 3 as per their order in manuscript.
Thank you very much! The order of tables was revised.
- Figure 2 has a very poor resolution. I encourage authors to add a better resolution figure 2.
Reviewer 3 recommends not to discuss the SNP rs1800797 in the manuscript, as both SNPs are in strong linkage disequilibrium. Therefore, this figure was removed from the manuscript.
We hope we have satisfied the concerns of the reviewer with regard to the points marked.
With kindest regards
Susanne Schulz on behalf of all authors

Reviewer 3 Report
Comments and Suggestions for Authors
The authors have generated a 10-year study that demonstrates IL-6, as well as rs1800795 and rs1800797 variants, has an increased risk for cardiovascular events. It was further determined that these CV events were independent from co-factors such as sex, age, BMI, smoking, etc. The benefit of this study is that the cardiovascular events include myocardial infarction, stroke/transient ischemic attack, cardiac death, death according to stroke, making it quite comprehensive. However, I have several questions regarding the linkage disequilibrium of the promoter SNPs rs1800795 and rs1800797. Please see my specific points of concern below:
1) It seems as though the rs1800797 polymorphism is always associated with rs1800795. Is it possible to distinguish the CVD risk of rs1800797 alone? In other words, it is possible to decouple CVD risk between both polymorphisms? Otherwise, how can you determine that rs1800797 increases CVD risk, it may just be due to the prevailing rs1800795 polymorphism.
2) Even if the rs1800797 resembles the results of rs1800795, I would still include rs1800797 in Table 2, Figure 3, and Table 4.
3) The conclusion only lists rs1800975 in the cardiovascular prognosis profile. If it is possible to show rs1800797 results alone, I would also include this in the conclusion. Otherwise, I would put less emphasis on rs1800797 throughout the text and remove it from the title.
Round 2
Reviewer 2 Report
Comments and Suggestions for Authors
The manuscript can be accepted in current form